# Machine-Learning Based Analysis of Liquid Water Path Adjustments to Aerosol Perturbations in Marine Boundary Layer Clouds Using Satellite Observations

Lukas Zipfel [1,2,*], Hendrik Andersen [1,2] and Jan Cermak [1,2]

1 Institute of Meteorology and Climate Research, Karlsruhe Institute of Technology (KIT), 76131 Karlsruhe, Germany; hendrik.andersen@kit.edu (H.A.); jan.cermak@kit.edu (J.C.)
2 Institute of Photogrammetry and Remote Sensing, Karlsruhe Institute of Technology (KIT), 76131 Karlsruhe, Germany
* Correspondence: lukas.zipfel@kit.edu

**Abstract:** Changes in marine boundary layer cloud (MBLC) radiative properties in response to aerosol perturbations are largely responsible for uncertainties in future climate predictions. In particular, the relationship between the cloud droplet number concentration ($N_d$, a proxy for aerosol) and the cloud liquid water path (LWP) remains challenging to quantify from observations. In this study, satellite observations from multiple polar-orbiting platforms for 2006–2011 are used in combination with atmospheric reanalysis data in a regional machine learning model to predict changes in LWP in MBLCs in the Southeast Atlantic. The impact of predictor variables on the model output is analysed using Shapley values as a technique of explainable machine learning. Within the machine learning model, precipitation fraction, cloud top height, and $N_d$ are identified as important cloud state predictors for LWP, with dynamical proxies and sea surface temperature (SST) being the most important environmental predictors. A positive nonlinear relationship between LWP and $N_d$ is found, with a weaker sensitivity at high cloud droplet concentrations. This relationship is found to be dependent on other predictors in the model: $N_d$–LWP sensitivity is higher in precipitating clouds and decreases with increasing SSTs.

**Keywords:** aerosol–cloud interactions; liquid water path; cloud droplet number concentration; machine learning; gradient boosting regression trees; marine boundary layer clouds; remote sensing; satellite observations

## 1. Introduction

Marine boundary layer clouds (MBLC) make up a large part of global cloud cover as they are persistently present over more than 20% of the Earth's oceans in the annual mean [1]. This is especially the case for the tropical and subtropical oceans off the west coasts of the continents, where semi-permanent stratocumulus sheets can cover more than 50% of the surface annually [1,2]. By reflecting solar radiation to a much greater degree than the ocean beneath, while only having a minor effect on the outgoing longwave radiation, these clouds play an important role in Earth's energy budget by exerting a large net cooling effect [1,3]. Therefore, even a comparatively small increase in the albedo of MBLC could offset part of the global warming due to increasing concentrations of greenhouse gases [4].

Stratocumulus cloud properties and their radiative characteristics, such as cloud albedo, horizontal and vertical extent, lifetime and precipitation susceptibility, are dependent on environmental conditions. Aerosols in their role as cloud condensation nuclei (CCN) affect the cloud albedo via changes in the cloud droplet number concentration ($N_d$), also known as the Twomey effect [5]. Subsequent cloud adjustments to aerosol perturbations may lead to changes in cloud fraction and in the liquid water path (LWP), further altering the radiative properties of the cloud. The magnitude and sign of the radiative

forcing due to these aerosol–cloud interactions remain among the largest uncertainties in projections of future climate [6–8]. While there is a large body of research on the Twomey effect from the last decades, e.g., [5,9–17], this is much less the case for LWP adjustments, where disagreement between observations and models is large and even the sign of the aerosol effect on LWP is unclear [18]. As LWP is the main controlling factor of liquid-cloud albedo [18], it is therefore important to better understand the effect of aerosols on LWP to ultimately improve climate model predictions.

Using the $N_d$–LWP relationship as a measure for the LWP adjustment of clouds to an aerosol perturbation, [18] outline several counteracting pathways of how aerosols can impact the LWP in MBLC. Small cloud droplets in situations with elevated $N_d$ can suppress the formation of precipitation, prolonging the lifetime of the cloud and increasing LWP [19]. Preliminary results of the CLoud–Aerosol–Radiation Interaction and Forcing: Year 2017 (CLARIFY-2017) and indications from the NASA ObseRvations of Aerosols above CLouds and their intEractionS (ORACLES) campaigns support this with findings of decreased drizzle formation in polluted (high $N_d$) clouds [20–22]. A second pathway that is hypothesized to increase LWP is the process of warm cloud invigoration, where the condensation and LWP build-up under low-$N_d$ conditions is limited by the small droplet surface area, which is then increased following an increase in $N_d$ [23]. On the other hand, the entrainment of relatively dry air at the cloud top leads to a decrease in LWP with increasing $N_d$ by way of faster evaporation of smaller cloud droplets [24–26]. The resulting evaporative cooling induces a (self-amplifying) positive feedback that enhances the entrainment–evaporation process, further decreasing LWP [27–29]. The extent to which these processes affect clouds under different meteorological conditions and cloud states is still unclear and previous estimates of the relationship between aerosols and LWP span from positive [23,30–34] to negative [24,25,29,35–37] with some studies showing a bidirectional relationship [38,39] as summarized in in Gryspeerdt et al. [18]. The recent study by Gryspeerdt et al. [18] found a possible explanation for these varying results in the non-linear relationship of $N_d$ and LWP, where the $N_d$–LWP relationship is positive at low $N_d$ and is reversed in high $N_d$ situations. A parameter that was found to influence this relationship is the state of the cloud (precipitating/non-precipitating). Precipitating clouds were shown to display an increase in LWP with increasing $N_d$ in the past [29,33]. SST is another important controlling factor for the $N_d$–LWP relationship [40] and radiative properties [41] in MBLCs. Higher SST leads to a deeper boundary layer with decreased stability and thus supports entrainment drying at the cloud top, thereby accelerating the evaporation at the cloud top.

Given the non-linear nature of the $N_d$–LWP relationship, and its potential dependence on meteorological factors and the cloud state, machine learning methods are ideally suited to analyze these processes in observational data sets. While machine learning techniques are becoming a popular tool in Earth sciences [42], and have been used to study aerosol–cloud interactions before [43–45], their potential to study LWP adjustment processes has not yet been explored.

The goal of this study is to improve the understanding of the $N_d$–LWP relationship in MBLC of the Southeast Atlantic region, specifically focusing on its dependency on meteorological conditions and cloud state. To this end, a large data set of polar-orbiting satellite observations and reanalysis data are analyzed in a machine learning framework.

## 2. Materials and Methods

This study is conducted for a 10° by 10° region in the Southeast Atlantic (0° E–10° E and 10° S–20° S), characterised by a high annual coverage of MBLC largely in the form of stratocumulus clouds [2]. The Southeast Atlantic has recently been the focus of multiple large aircraft campaigns to better understand interactions of stratocumulus clouds with the seasonally occurring biomass burning aerosol layer above the cloud deck [20,21,46]. Here, the focus is not explicitly on the effects of biomass burning aerosols on clouds, but rather on the LWP adjustment of stratocumulus clouds to aerosol perturbations, using $N_d$ as a mediating variable [18]. A combination of observation data from multiple satellite

sensors and reanalysis model output is used to create a data set for the period of July 2006–April 2011, which is then utilized to train a machine learning algorithm to predict LWP. An overview of the variables used in this work is shown in Table 1.

**Table 1.** Overview of the variables used in the machine learning model.

| Variable Name | Abbreviation | Origin |
|---|---|---|
| **Predictors** | | |
| Temperature below cloud | $T_{bc}$ | ERA5 |
| Vertical velocity below cloud | $w_{bc}$ | ERA5 |
| Winds below cloud | $u_{bc}$ / $v_{bc}$ | ERA5 |
| Winds above cloud | $u_{ac}$ / $v_{ac}$ | ERA5 |
| Relative humidity below cloud | $RH_{bc}$ | ERA5 |
| Relative humidity above cloud | $RH_{ac}$ | ERA5 |
| Mean sea level pressure | MSL | ERA5 |
| Sea surface temperature | SST | ERA5 |
| Estimated inversion strength | EIS | ERA5 |
| Cloud top height | CTH | CALIPSO |
| Precipitation fraction | PF | CloudSat |
| Cloud droplet number concentration | $N_d$ | MODIS |
| **Predictand** | | |
| Liquid water path | LWP | AMSR-E |

*2.1. Data*

Observation data on cloud properties are taken from the CALIPSO-CloudSat-CERES-MODIS Merged Release B1 (C3M) product. The C3M data set aggregates observations from multiple sensors (Cloud-Aerosol Lidar and Infrared Pathfinder Satellite Observation (CALIPSO), Cloudsat, Clouds and the Earth's Radiant Energy System (CERES) and Moderate Resolution Imaging Spectroradiometer (MODIS)) on CERES footprints with a resolution of ∼20 km [47]. In the C3M data set, each CERES footprint represents an individual observation with data from all four instruments collocated. To account for the higher resolution and the vertical profiles provided by the original CALIPSO and CloudSat products, C3M contains a maximum of 16 cloud groups and a maximum of six cloud layers per CERES footprint. Individual cloud groups (i.e., clouds within the same CERES Footprint that are separated by clear sky areas) are distinguished as seen from above while the cloud layers are distinguished vertically. In order to filter for low-level clouds and to exclude the influence of additional cloud layers above, only observations of single-layer clouds with a cloud top height (CTH) below 3 km [48] as detected by CALIPSO are used. CTH is defined as the median of all cloud groups in a CERES footprint. To inform the machine-learning model about possible precipitation, the CloudSat precipitation flag is used to calculate the precipitation fraction (PF). The PF is defined as the number of cloud groups where precipitation is detected by CloudSat (precipitation classes can either be liquid, solid or drizzle), divided by the total number of cloud groups for each CERES footprint. The large majority (>99%) of precipitating clouds in this data set are classified as drizzle with no instances of solid precipitation detected.

The cloud-droplet number concentration ($N_d$) is calculated using MODIS retrievals of the effective cloud-droplet radius ($r_e$), the cloud optical depth ($\tau_c$), the cloud-top temperature and the cloud-top pressure according to Grosvenor et al. [49]:

$$N_d = \frac{\sqrt{5}}{2\pi k} \sqrt{\frac{f_{ad}\, c_w\, \tau_c}{Q_{ext}\, \rho_w\, r_e^5}} \qquad (1)$$

with $k = 0.8$, $f_{ad} = 0.66$ and $Q_{ext} = 2$. The overall uncertainty in the calculated $N_d$ due to the cumulative uncertainties in the retrievals of $r_e$ and $\tau_c$ are estimated to amount to around 78% [49].

Meteorological reanalysis data are taken from the ERA5 dataset provided by the European Centre for Medium-Range Weather Forecasts (ECMWF) at a 0.25° × 0.25° resolution on an hourly basis [50,51]. Data for air temperature, relative humidity (RH), vertical velocity, u and v wind components, sea-surface temperature (SST) and mean sea-level pressure (MSL) are collocated with the C3M data. For each CERES footprint the cloud-base height (CBH) from CALIPSO is used to select the nearest pressure level below the cloud for vertically resolved ERA5 variables (temperature, RH and winds). Additionally, CTH is used to select RH and winds at the nearest pressure level above the cloud. Finally, the estimated inversion strength is calculated according to Wood and Bretherton [52] using the 2 m air temperature and the temperature at 700 hPa from ERA5 and assuming a surface pressure of 1010 hPa.

The liquid-water path (LWP) is obtained from the Level-2B precipitation product Version 3 of the Advanced Microwave Scanning Radiometer-Earth Observing System (AMSR-E) sensor aboard the Aqua satellite. This data set is independent of the MODIS-derived $N_d$ and is utilized to eliminate the potential risk of introducing a pseudo-relationship through correlated errors in $N_d$ and LWP retrievals. The data have a resolution of 5 km across track and 10 km along track [53]. To achieve a comparable spatial resolution to the CERES footprints, the five pixels nearest to the center of each CERES footprint are selected to calculate the mean LWP for that footprint. One drawback of the AMSR-E LWP is that for low LWP cases, cloud water cannot be separated from rainwater (mostly drizzle in this data set), meaning that in such situations, AMSR-E LWP is biased high when compared to other products [54].

Spatial gradients are inherent in the LWP and many predictors in the data of the Southeast Atlantic that are used here. However, these are not necessarily directly linked to the $N_d$–LWP relationship. To ensure that the model is not primarily exploiting the spatial gradients to predict the LWP, a secondary data set is created by removing the spatial gradients (Figure S1). The results based on this anomalous data set are shown in Figures S2–S5 in the Supplementary.

*2.2. Methods*

Gradient-boosting regression trees (GBRTs) are used to model the LWP using the set of 12 predictor variables (model features) described in Table 1. GBRTs are a robust machine-learning technique that features the advantages of tree-based methods, which allow for the use of different data types (e.g., categorial or numerical data) and do not make/need prior assumptions concerning the distribution of the data. They are capable of representing and quantifying complex non-linear relationships, while considering interactive effects between the predictors [55]. GBRTs have been successfully applied to study aerosols and clouds in the past (e.g., [44,56–58]). As a result of uncertainties in the retrieval of $r_e$ and $\tau_c$, unrealistic values for $N_d$ may be calculated. Therefore, only observations where $N_d$ is within percentiles 1–99 are used for the analysis to remove extreme values in $N_d$, yielding N = 29,901 observations. To evaluate the model, the data set is randomly split into training (70%, N = 20,931) and test data sets (30%, N = 8970). To find the best set of hyperparameters, multiple instances of the model are first run with manually selected parameters to find a grid of the most suitable settings. In a second step, the final hyperparameters are chosen by applying a grid search approach (Table 2).

**Table 2.** Overview of the hyperparameters used in the grid search approach. Values in bold are used in the GBRT model. Parameters not listed here are kept in default configuration.

| Hyperparameter | Value | | | | |
|---|---|---|---|---|---|
| n_estimators | 600 | 800 | **1000** | 1500 | 2000 |
| learning_rate | **0.01** | 0.05 | 0.1 | 0.25 | 0.5 |
| max_depth | 1 | 3 | **5** | 7 | 10 |
| min_samples_leaf | 1 | 15 | **50** | 80 | 180 |

In order to interpret the model predictions and to analyse the $N_d$–LWP relationship, an explainable machine learning tool, the Shapley additive explanation (SHAP) values, are used. SHAP values quantify the contributions of each model feature to each individual (local) model prediction [59,60]. An example for a single set of observations is provided in Figure 1. SHAP values retain local accuracy, so that each individual model prediction is equal to the sum of the SHAP values of all features and the mean model prediction. Following from this, individual SHAP values can be positive (increase in the model prediction due to the specific feature value) or negative (decrease in the model prediction), and are calculated for each individual feature and each individual model prediction. Figure 1 shows that the observed $N_d = 10.041$ contributes a SHAP value of $-8.41$ to the local prediction. Since SHAP values are provided in units of the predictand, this translates to a decrease in the LWP prediction by $-8.41$ gm$^{-2}$. Accordingly, the mean absolute SHAP value of a feature directly indicates the strength of the influence of this feature on the model prediction. Interactive effects between the predictors are quantified as the change in the contribution of a feature to the model prediction depending on the presence/absence of a second feature (SHAP interaction values). For additional information on the theoretical background and technical details of SHAP values, the reader is referred to Lundberg et al. [59,60]. SHAP values have seen use in the field of medical science [61] and more recently also in the environmental sciences [62].

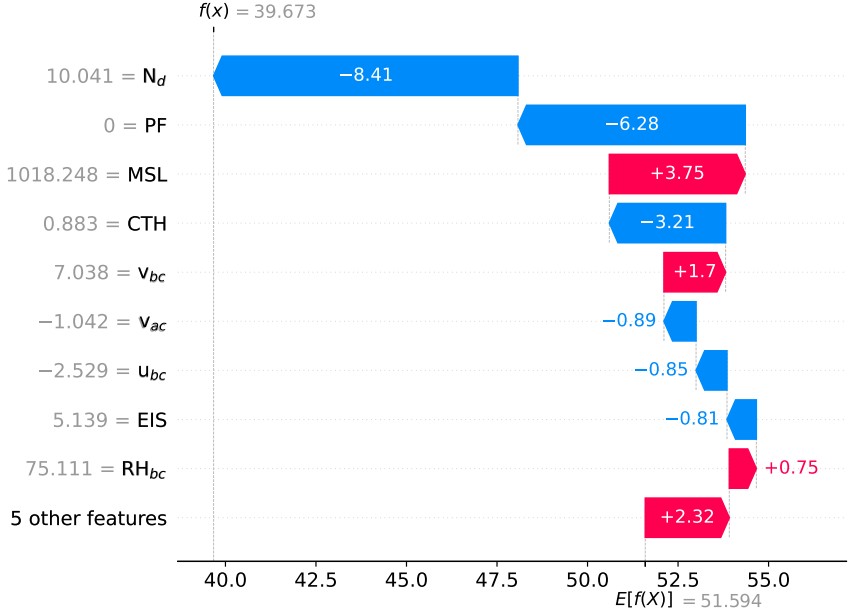

**Figure 1.** Exemplary plot for a single set of observations (one CERES footprint) showing the individual contributions (SHAP values) of each feature to the (local) prediction of LWP. Red arrows represent positive SHAP values (increasing the prediction), negative SHAP values are blue (decreasing the prediction). The feature values are displayed in grey. The mean model prediction is displayed as $E[f(X)] = 51.594$. The sum of all SHAP values and the mean model prediction result in the final prediction for this set of observations $f(x) = 39.673$.

## 3. Results and Discussion

Figure 2a shows the correlation between $N_d$ and LWP at the spatial scale of the C3M data, summarizing all data points in $1° \times 1°$ grid boxes in the study domain. Mostly positive correlations between $N_d$ and LWP are found in the data sets analyzed here. This is particularly the case for low $N_d$ values (<30, Figure 2b), where the average weighted correlation (weighted by number of observations per pixel) is 0.14. However, when considering only observations with $N_d > 30$, the correlation decreases overall (average weighted correlation: 0.05, Figure 2c), with negative correlations apparent in southwestern parts of the study domain. To account for this non-linearity in the $N_d$–LWP relationship, and to analyze its dependence on meteorological factors and cloud state, a machine learning model is used in the following. This approach allows for the estimation of sensitivities of LWP to perturbations in $N_d$, including interaction effects with secondary parameters, that can be compared to other studies.

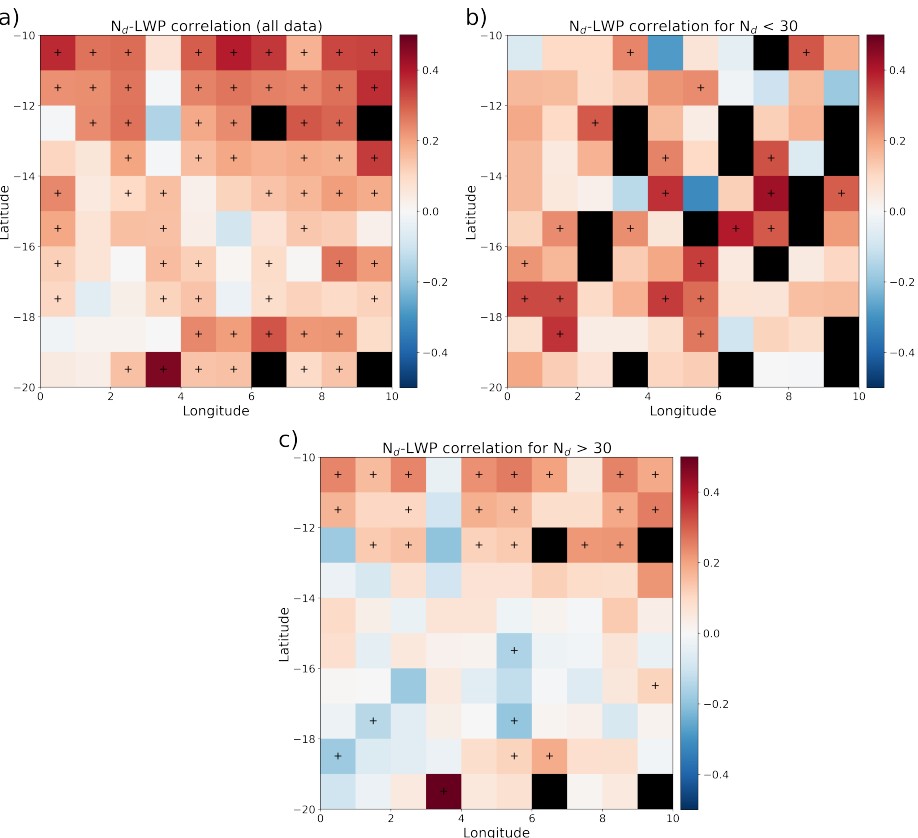

**Figure 2.** Pearson correlation between $N_d$ and LWP for the data aggregated in $1°$ by $1°$ pixels for the study region. A '+' marks significant correlations ($p < 0.05$). Pixels with less than 20 observations are excluded (black). The three panels show (**a**) all observations, (**b**) only observations with $N_d < 30$ and (**c**) observations with $N_d > 30$.

A comparison of the model based on the data set with spatial gradients removed (Figures S2–S5), and the model based on the observations, reveals no significant differences with respect to the SHAP values and interactive effects. This indicates that the model performance is not negatively impacted by the spatial gradients present in the data set. Therefore, only the model based on the untreated observations is discussed in the following. The GBRT model is able to explain 70% of the variability in the LWP ($R^2$ of the independent test data = 0.70). An overview of the importance and the contributions of each predictor to the model-predicted LWP is shown in Figure 3. The predictors are sorted by their mean absolute SHAP value as a measure of importance with respect to the prediction of LWP. PF, CTH, MSL and $N_d$ are identified as the most important variables. The fact

that cloud state variables are among the most important features in the model is to be expected, given that they are the result of the prevailing environmental conditions, but also more directly related to the predicted LWP. As such, the importance of the two groups (cloud state variables and environmental conditions) should not be directly compared. In Figure 3, each data point is represented by a dot for each of the predictors with the associated SHAP value (contribution to the predicted LWP) on the horizontal axis, while the normalized predictor value is indicated by the coloring (ranging from the 5th to the 95th percentile), where blue shows a below-average feature value, and red an above-average feature value. A positive sensitivity of LWP to the three most important features PF, CTH and MSL is apparent. This relationship for PF is to be expected, as only clouds with a sufficiently high LWP form precipitation, but may also be amplified by the inability of AMSR-E to distinguish between cloud water and rain water for low LWP values, meaning that the LWP may contain drizzle/precipitation water in cases where $PF > 0$. Similarly, a higher CTH in stratocumulus clouds is linked to a deeper boundary layer and thicker MBLCs [43,63]. The importance of MSL for the model predictions of LWP underscores the role of dynamics for MBLCs in the Southeast Atlantic, as suggested by [57,64]. Furthermore, Figure 3 suggests an overall positive $N_d$–LWP relationship that is in agreement with the positive correlations shown in Figure 2. In the following, this $N_d$–LWP relationship and its dependence on meteorological factors and cloud state is explored in detail.

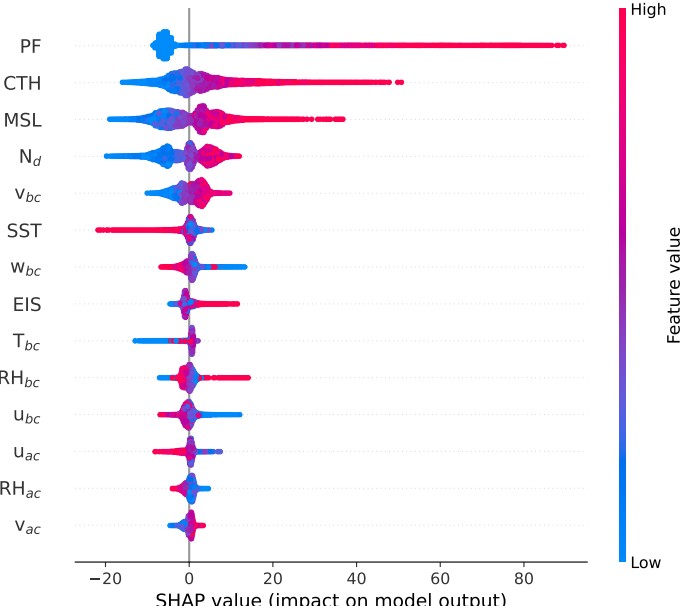

**Figure 3.** "Beeswarm" plot showing the impact of each feature on the model prediction. Every dot represents an observation with color signifying the original value and the corresponding SHAP value shown on the horizontal axis. Density is made visible by stacking dots on the vertical. The features are sorted by their mean absolute SHAP values in descending order.

The main effects of $N_d$ on LWP are shown in Figure 4. Main effects are equivalent to the fraction of the SHAP values that are solely attributable to a single feature with all interaction effects removed. A positive relationship of $N_d$ and LWP is found for low $N_d$ values. The relationship is much weaker for higher $N_d$, where the predicted LWP is less sensitive to a further increase in droplet concentration, underscoring the nonlinearity of the $N_d$–LWP relationship. This is partly in agreement with [18], who also found a positive sensitivity of LWP to $N_d$ at lower $N_d$. However, the negative $N_d$–LWP relationship at higher $N_d$ values found in Gryspeerdt et al. [18] is not apparent in the data set analyzed here. A potential cause for this difference could be the differences in spatial resolution and specific cloud filters in both studies, which have been shown to lead to systematic biases [49]. In particular, the spatial and temporal aggregation of a data set can have a

significant impact on the statistical relationship between $N_d$ and LWP, and even lead to a change in its sign when using Level 3-type data [65], as in Gryspeerdt et al. [18]. Another aspect could be the different method used to calculate the $N_d$, as Gryspeerdt et al. [18] use the method from Bennartz and Rausch [66], whereas here, we use the method presented in Grosvenor et al. [49]. In the study domain, however, the retrieved $N_d$ agrees closely for both methods [49]. It should be noted that this finding of a nonlinear $N_d$–LWP relationship that saturates at higher cloud droplet concentrations agrees well with previous modeling work done in the Southeast Pacific with a regionally-nested configuration of the Met Office Unified Model, that has found a very similar relationship [32].

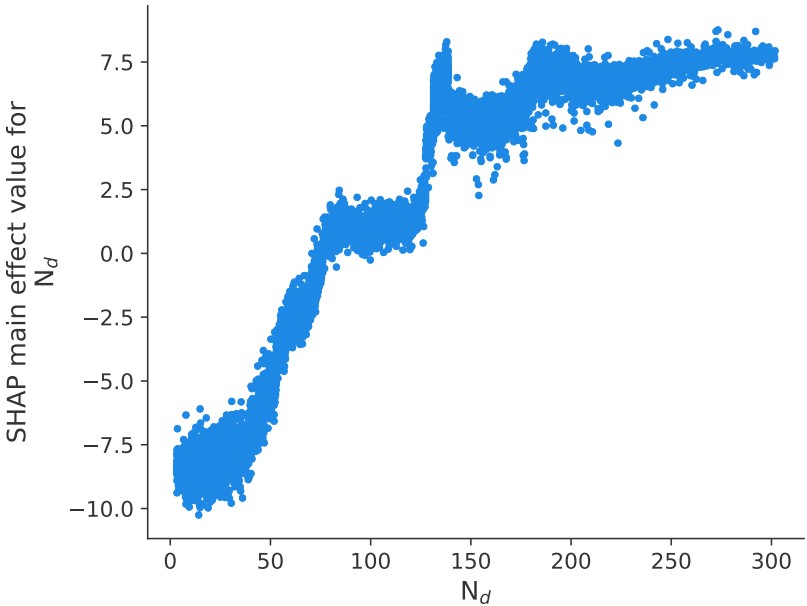

**Figure 4.** The main effects of $N_d$ on the prediction of LWP. Main effects show the changes in the model prediction that are solely attributed to the corresponding observed $N_d$ anomaly by removing the interaction effects with other model features from the SHAP values.

Figure 5 shows the influence of precipitation fraction on the $N_d$–LWP relationship. The SHAP values for $N_d$ are shown in Figure 5a, including interactive effects (in contrast to Figure 4). The observed PF that corresponds to each data point is indicated by color, increasing from blue to red. Overall, a higher variability in $N_d$ SHAP values is found when interaction effects are taken into account, compared to only the main effects of $N_d$ (Figure 4). This can largely be attributed to the importance of the interaction effects between $N_d$ and PF (Figure 5b). The positive slope of the interaction effects of PF on the $N_d$–LWP relationship suggests that the model actively uses the information of PF to improve the model performance. This indicates that the strength of the positive $N_d$–LWP relationship, which is assumed to be related to precipitation suppression, is amplified in conditions that already partially develop drizzle. Preliminary results from the CLARIFY-2017 in-situ measurements support this finding with higher LWP and a lower amount of drizzle formation in clouds with elevated $N_d$ [20]. A second microphysical process that could explain the amplified positive $N_d$–LWP relationship in cases of drizzle could be related to the simultaneous removal of cloud water and droplets. Since drizzle acts as a sink for both $N_d$ and LWP at the same time, this could increase the strength of their positive relationship [67]. The dependency of the LWP response to aerosol/increasing droplet concentrations on precipitation is also in agreement with previous studies that find a positive relationship between $N_d$ and LWP in precipitating conditions [29,33].

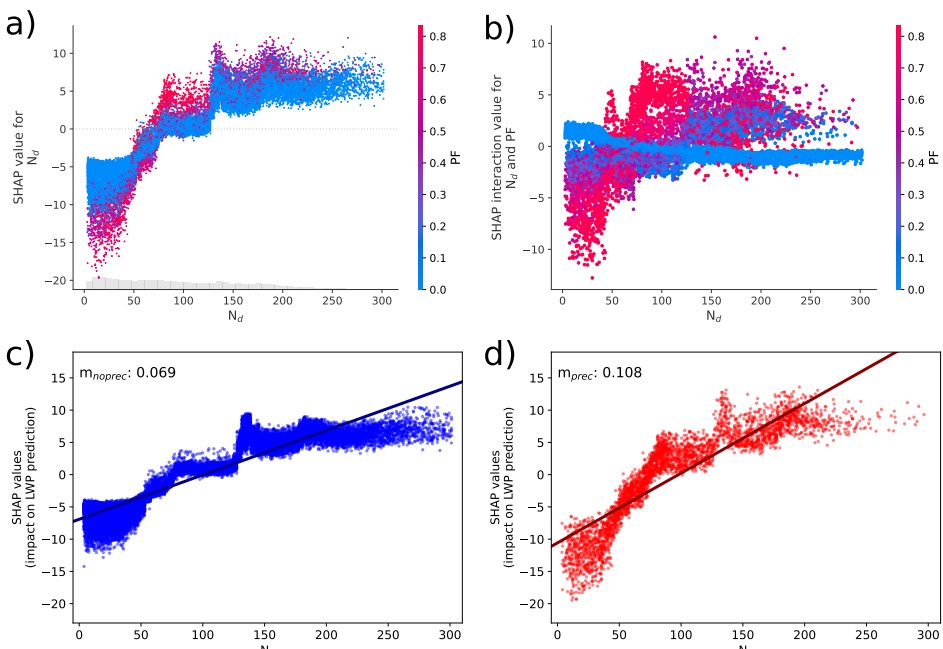

**Figure 5.** Influence of the PF on the $N_d$–LWP relationship. SHAP values (**a**) and interaction effects (**b**) for $N_d$ with the color showing the PF. The lower panels show the SHAP values for $N_d$ for non-precipitating situations ($PF = 0$, (**c**)) and for situations where at least 50% of the cloud groups precipitate ($PF \geq 0.5$, (**d**)). $m_{noprecip}$ and $m_{precip}$ are defined as the slope of the linear regression for the corresponding subset of $N_d$ SHAP values.

The lower panels of Figure 5 show a comparison of the $N_d$–LWP sensitivity for non-precipitating ($m_{noprecip}$, Figure 5c) and precipitating clouds ($m_{precip}$, Figure 5d) as the slope of the linear regression for $N_d$ SHAP values. While the finding of a stronger $N_d$–LWP relationship at low $N_d$ and a weakening at higher $N_d$ is consistent across both subsets, precipitating clouds show a markedly higher sensitivity of LWP to changes in $N_d$ ($m_{precip} = 0.108$) compared to non-precipitating clouds ($m_{noprecip} = 0.069$).

Figure 6a shows the influence of $N_d$ on LWP predictions, with color showing SST (same as in Figure 5a for PF). The differences in LWP sensitivity for low (blue dots) and high (red dots) SST suggest only a minor direct influence of SST on the $N_d$–LWP relationship. The details of this influence are shown in Figure 6b as SHAP interactive effects. High (low) values for SST weaken (enhance) the sensitivity of LWP to changes in $N_d$. This effect seems to be more pronounced in situations with a lower amount of cloud droplets. Higher SST conditions are associated with increased surface fluxes, a deeper and less stable marine boundary layer, and an increased moisture difference between the marine boundary layer and the troposphere [68,69]. In these conditions, the evaporation-entrainment process may be facilitated [40], leading to a less positive $N_d$–LWP relationship. This finding is in good agreement with recent studies by Zhou et al. [41] and Zhang et al. [40]. It has to be noted, that Figure 6b shows the direct impact of SST on the $N_d$–LWP relationship in MBLC. However, since SSTs are known to be drivers for other meteorological factors and the cloud state variables as described above, they may also have an indirect impact on the $N_d$–LWP relationship exerted through these features in the model. Compared to other studies, this approach has the potential to separate the direct influence of SST from the indirect effects exerted through changes in secondary features like CTH, LTS and RH, as these are included in the model.

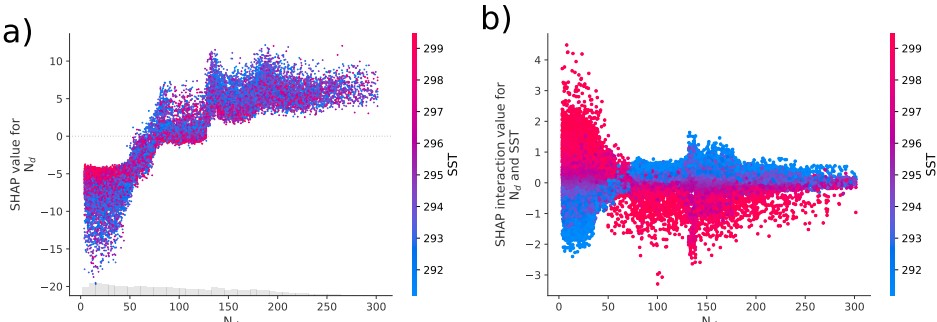

**Figure 6.** Influence of SST on the $N_d$–LWP relationship. Panels (**a**,**b**) are the same as in Figure 5 with color showing SST.

## 4. Conclusions

In this work, a machine learning model was trained on observation data and reanalysis model output representing important parameters in cloud development to predict the LWP of MBLC in the Southeast Atlantic. The goal of this study was to improve the understanding of how changes in $N_d$ affect the LWP of MBLC and how this relationship is influenced by meteorological factors and cloud state. The main findings of the study are that

- Within the machine-learning model, the most important cloud state parameters for the prediction of LWP are PF, CTH, and $N_d$, while the most important environmental predictors are MSL, $v_{bc}$ and SST. The machine-learning model is able to explain 70% of the observed variability in LWP ($R^2 = 0.70$).
- Overall, a nonlinear but positive sensitivity of LWP to changes in $N_d$ is found, with a positive relationship at low $N_d$ values, which saturates at higher $N_d$ values. Unlike findings in a previous global study [18], the $N_d$–LWP relationship at higher $N_d$ is not negative in the data set used here for the Southeast Atlantic.
- Marked differences are found in the sensitivity of LWP to changes in $N_d$ for precipitating and non-precipitating cloud groups. The stronger sensitivity is likely due to an amplified importance of precipitation suppression in situations that already develop some drizzle.
- Changes in SST show a direct influence on the $N_d$–LWP relationship, with a decreased sensitivity of LWP to $N_d$ at higher SSTs. This may be attributed to increased evaporation-entrainment and deeper clouds due to the lower stability at higher SSTs.

**Supplementary Materials:** The following are available online at https://www.mdpi.com/article/10.3390/atmos13040586/s1, Figure S1: Calculation of the spatial anomalies, Figures S2–S5: Results of the GBRT model based on the data set with the spatial gradients removed.

**Author Contributions:** Conceptualization, L.Z., H.A. and J.C.; methodology, L.Z. and H.A.; software, L.Z.; formal analysis, L.Z.; writing—original draft preparation, L.Z.; writing—review and editing, H.A. and J.C.; visualization, L.Z.; funding acquisition, H.A. and J.C. All authors have read and agreed to the published version of the manuscript.

**Funding:** This work has received funding from the European Union's Horizon 2020 research and innovation program under grant agreement no. 821205 (FORCeS).

**Data Availability Statement:** Publicly available data sets were analyzed in this study. The C3M data set can be found at the NASA Langley Research Center: https://doi.org/10.5067/AQUA/CERES/NEWS_CCCM-FM3-MODIS-CAL-CS_L2.RELB1 (accessed on 12 February 2020). ERA5 data on single levels and on pressure levels are available at the Copernicus Climate Change Service (C3S) Climate Date Store: Single levels—https://doi.org/10.24381/cds.adbb2d47 (accessed on 8 March 2021), Pressure levels—https://doi.org/10.24381/cds.bd0915c6 (accessed on 8 March 2021). The AMSR-E data set is available at the NASA National Snow and Ice Data Center: https://doi.org/10.5067/AMSR-E/AE_RAIN.003 (accessed on 25 September 2021).

**Acknowledgments:** The authors would like to thank Daniel Grosvenor for his help regarding the calculation of $N_d$ and Robert Wood for providing insights into his algorithm to calculate EIS. The authors further express their thanks to three anonymous reviewers for their careful and constructive comments which have helped improve the manuscript.

**Conflicts of Interest:** The authors declare no conflict of interest.

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
