# Peer review of "Machine-Learning Based Analysis of Liquid Water Path Adjustments to Aerosol Perturbations in Marine Boundary Layer Clouds Using Satellite Observations"

_atmosphere, doi:10.3390/atmos13040586_

Round 1

Reviewer 1 Report

Attached

Reviewer 2 Report

Summary: The paper uses machine learning to present the non-linear relationship between Nd (proxy for aerosols) and Liquid Water Path. Satellite and reanalysis data products are used to generate the dataset for training. A gradient boosted regression model is used to create a model for LWP from 11 input variables. The contribution of each feature to the deviation from the model mean is analyzed using Shapley values. A positive relationship for SHAP contribution is predicted between Nd-LWP for smaller values of Nd, almost saturating at higher values of Nd. The interaction SHAP values are also looked at and the results indicate that precipitation fraction (PF) may be a major feature in defining the Nd-LWP relationship. The article is well written and presents a logical train of thought, with a few exceptions identified below. The results are not surprising, but they serve to demonstrate the method. There is one fundamental concern that needs to be clarified before the paper can be published. A number of other recommendations are also made. 

Major issues: The authors remove the lat-lon based gradient of LWP during data preparation to avoid bias and prevent it from being the dominating factor in the GBRT model. Precipitation fraction can be thought to be similar to this (as higher PF corresponds with higher LWP) where it can bias the model to be the most important feature. This is seen from the SHAP values where its highest value is more than 4 times that of Nd. Similar argument can also be extended for CTH. This may suggest that the model, though captures the observation to an extent may be biased based on PF or CTH and the effect of other inputs not fully captured. The authors should further discuss why certain factors are considered for removal, whereas other factors are included in the analysis, and how that influences the results.

Minor issues:

Section 2.1 Data- Lines 93-105: It is not fully clear what is meant by a cloud group and how the precipitation fraction is calculated. What constitutes liquid, solid and drizzle cloud groups? This section needs to be made clearer to those who are not regularly using satellite data.

Section 2.1 Data- lines 106-108: It might be good to mention the uncertainty range for Nd values calculated with this method.

Section 2.1 Data- line 127: It is worth mentioning why spatial gradients are inherent in the LWP for the chosen region.

Section 2.2 Methods- Line 155: Doesn’t the SHAP value quantify the contribution of a feature to the deviation of a local model prediction from the global(mean) model prediction? This is put differently in lines 159-160. It would be worth clarifying that.  

Section 3 Results and Discussion- line 177: It is stated that the GBRT model is able to explain nearly 70 percent of the variability in the LWP anomalies. This statement is vague because no information is provided on how this value is estimated. Please provide details.

Section 3 Results and Discussion- lines 203- 213: Could it be possible that the Nd-LWP relationship at higher Nd values is not fully captured if the model was biased by PF and CTH. If they are the major contributing features for the model could they mask the effects of other input features that could influence entrainment and the negative Nd-LWP relationship at higher Nd values?

Section 3 Results and Discussion- The entire SHAP value matrix (average of all) can help understand the mean interaction effects of input variable pairs and confirm that known relationships among the input variables are not violated. More information on this aspect should be provided.

Section 3 Results and Discussion- Figure 3: Shouldn’t higher values of RHac reduce the effect of entrainment drying on LWP? The SHAP values give a negative contribution.     

Section 3 Results and Discussion- Figure 6: Higher SST may be associated with deeper and less stable MBL. Large SST might help deepen the MBL so that LCL with respect to CTH is lower and may explain the positive SHAP contribution for small Nd values. And this should turn to negative for larger Nd anomalies due to entrainment effects being the major factor. But what might be the reasoning for the positive SHAP contribution for cluster of points for the highest Nd anomaly (large SST case)?

Reviewer 3 Report

Comment on “Machine-learning Based Analysis of Liquid Water Path Adjustments to Aerosol Perturbations in Marine Boundary Layer Clouds Using Satellite Observations” by Zipfel et al.

The manuscript presents the research on the LWP adjustments to aerosol perturbation using a machine learning method. The topic is important since it helps to improve our understanding of the aerosol impact on the radiative budget of the atmosphere. The method of the manuscript is machine learning which has been the trend in research since the last several years. I am in favour for the paper to be accepted to publish in the journal after minor revision.

Main points:

1. The introduction should have some in-situ observational studies. This is important since in-situ observations provide informations close to truth. A concern is the interpretation of the results using satellite and model only. Please see specific points.

2. Are you sure that all clouds in the box are stratocumulus? The region is in the transition zone from stratocumulus to cumulus. It is very possible that some cumulus clouds exist.

3. Table lists the variables used in the machine learning model. Only sea level pressure is used to represent the synoptic conditions. In fact, the wind direction and speed at levels above the boundary layer top are important for different aerosol scenarios (Haywood et al., 2021).

4. I have a question about the spatial gradient estimation (L127-136). You used a linear regression model to remove the background to get the anomalies. I wonder whether it was conducted at each pixel. I think it is better to use the Multivariate Polynomial Regression to get the background values on a 2D surface. In this way, the spatial gradients should represent the real spatial shape of the background. If it was conducted at each pixel, it may overestimate the gradient.

Specific points:

L19-20: Need a citation. Of which paper 20% cloud coverage is from?

L27: Mask? Probably use another word. For example, offset.

L32: “this perturbation”? Please specific. Perturbation in what?

L39: “As LWP is the main controlling factor of liquid-cloud albedo”. Albedo is a function of both Nd and LWP.

L60-61: “Precipitating clouds were shown to display an increase in LWP with increasing Nd in the past [26,30].”

L62-64: SST affects not only stability but also the height and structure of the boundary layer.

L118: “Finally, the estimated inversion strength is calculated according to [51] using the 2m air temperature from ERA5”. The estimated inversion strength in their original paper depends on several parameters

Why only 2m temperature was mentioned in the manuscript?

L209-213: “It should be noted though that this finding of a nonlinear Nd-LWP relationship that saturates at higher cloud droplet concentrations agrees well with previous modeling work done in the Southeast Pacific with a regionally-nested configuration of the Met Office Unified Model that has found a very similar relationship”.

Modelling results may or may not be correct and cannot be used to verify the satellite based analysis. It is important to use the in-situ observations. This is what is said in the main point 1 above. In fact, there have been some observational studies of the Nd-LWP, for example, Cui et al. (2014) which shows the aerosol & drop number concentrations and the LWP along 20 °S.

L224-227: “This dependency of the LWP response to aerosol/increasing droplet concentrations on precipitation is in agreement with previous studies that find a positive relationship between Nd and LWP in precipitating conditions [Chen et al., 2014; Neubauer et al., 2017].”

The cited paper were on satellite data and model result only. Again, see the main point 1.

One of the most important features of the aerosol and precipitation relationship is the enhanced warm rain process at low aerosol concentrations. Precipitation develops at low aerosol (drop) number concentrations. The aerosol particles are scavenged. There are not enough new aerosol to support the clouds. Therefore, LWP is low (Cui et al., 2014). The manuscript lacks such interpretation of microphysical processes.

L240-246: “Higher SST conditions are associated with increased surface fluxes, a deeper and less stable marine boundary layer, and an increased moisture difference between the marine boundary layer and the troposphere [64,65]. In these conditions, the evaporation-entrainment process may be facilitated [37], leading to a less positive Nd-LWP relationship. This finding is in good agreement with recent studies by Zhou et al. [38] and Zhang et al. [37].”

Higher SST conditions also cause the changes in the vertical structure within the boundary layer. It is common that the boundary layer becomes decoupled at high SSTs.

References

Chen, Y.C.; Christensen, M.W.; Stephens, G.L.; Seinfeld, J.H. Satellite-based estimate of global aerosol–cloud radiative forcing by marine warm clouds. Nature Geoscience 2014, 7, 643.

Neubauer, D.; Christensen, M.W.; Poulsen, C.A.; Lohmann, U. Unveiling aerosol–cloud interactions – Part 2: Minimising the effects of aerosol swelling and wet scavenging in ECHAM6-HAM2 for comparison to satellite data. Atmospheric Chemistry and Physics 2017, 17, 13165–13185. doi:10.5194/acp-17-13165-2017.

Cui, Z., Gadian, A., Blyth, A., Crosier, J., and Crawford, I.: Observations of the variation in aerosol and cloud microphysics along the 20∘ S transect on 13 November 2008 during VOCALS-REx, J. Atmos. Sci., 71, 2927–2943, https://doi.org/10.1175/JAS-D-13-0245.1, 2014.

Haywood et al. The CLoud–Aerosol–Radiation Interaction and Forcing: Year 2017 (CLARIFY-2017) measurement campaign. Atmospheric Chemistry and Physics 2021, 21, 1049–1084. doi:10.5194/acp-21-1049-2021.

Round 2

Reviewer 1 Report

The authors have nicely addressed my concerns in the text. My remaining concern is the #2 of the main comments in my first report. I still think the authors should separate the impact of cloud state from that of environmental conditions in the abstract and in the conclusion even though they are looking at direct impact. 

Author Response

Thank you for the quick reply. We have edited the Abstract and the Conclusions sections and separated the impact of the cloud state and the environmental variables as suggested.